# Low-Temperature Storage Improves the Over-Time Stability of Implantable Glucose and Lactate Biosensors

**DOI:** 10.3390/s19020422

**Published:** 2019-01-21

**Authors:** Giulia Puggioni, Giammario Calia, Paola Arrigo, Andrea Bacciu, Gianfranco Bazzu, Rossana Migheli, Silvia Fancello, Pier Andrea Serra, Gaia Rocchitta

**Affiliations:** Medical, Surgical and Experimental Sciences, University of Sassari, Viale San Pietro 43/b, 07100 Sassari, Italy; giuliamariagrazia@gmail.com (G.P.); gmcalia@uniss.it (G.C.); pa1989@live.it (P.A.); andreabacciu90@gmail.com (A.B.); gbazzu@uniss.it (G.B.); rmigheli@uniss.it (R.M.); sfancello@uniss.it (S.F.); grocchitta@uniss.it (G.R.)

**Keywords:** glucose, lactate, biosensors, low-temperature storage, over-time stability

## Abstract

Molecular biomarkers are very important in biology, biotechnology and even in medicine, but it is quite hard to convert biology-related signals into measurable data. For this purpose, amperometric biosensors have proven to be particularly suitable because of their specificity and sensitivity. The operation and shelf stability of the biosensor are quite important features, and storage procedures therefore play an important role in preserving the performance of the biosensors. In the present study two different designs for both glucose and lactate biosensor, differing only in regards to the containment net, represented by polyurethane or glutharaldehyde, were studied under different storage conditions (+4, −20 and −80 °C) and monitored over a period of 120 days, in order to evaluate the variations of kinetic parameters, as V_MAX_ and K_M_, and LRS as the analytical parameter. Surprisingly, the storage at −80 °C yielded the best results because of an unexpected and, most of all, long-lasting increase of V_MAX_ and LRS, denoting an interesting improvement in enzyme performances and stability over time. The present study aimed to also evaluate the impact of a short-period storage in dry ice on biosensor performances, in order to simulate a hypothetical preparation-conservation-shipment condition.

## 1. Introduction

Amperometric biosensors, a category of chemical sensors, combine the specificity of biological recognition procedures with high sensitivity [1], linked also to the validity of electrochemical techniques. Biosensors hold a biological element, represented by tissues or receptors, nucleic acids, antibodies, proteins or even whole cells, but more often enzymes, for the selective recognition of the studied compounds. In the case of amperometric biosensors, an enzymatic reaction usually generates an electrical signal that is proportional to the studied-compound concentration [1,2]. In this study, two different biosensors were studied. The first one was a glucose biosensor that exploits the capability of glucose oxidase (GOx) to selectively convert D-glucose as follows:β-D-Glucose + FAD^+^-GOx → D-Glucono-δ-Lactone + FADH_2_-GOx
FADH_2_- GOx + O_2_ → FAD^+^-GOx + H_2_O_2_

The second was a lactate biosensor that employs the capability of lactate oxidase (LOx) to selectively transform L-lactate as follows:L-Lactate + FAD^+^-LOx → Pyruvate + FADH_2_-LOx
FADH_2_-LOx + O_2_ → FAD^+^-LOx + H_2_O_2_

The byproduct H_2_O_2_ is easily oxidized over a platinum electrode by applying an anodic potential of +700 mV vs. Ag/AgCl [3] as follows:H_2_O_2_ → O_2_ + 2H^+^ + 2e^−^

The obtained current is related to the glucose or lactate concentrations [4,5,6,7,8,9,10].

Among the several issues concerning biosensing, one of a certain importance is the stability of the biosensor performances. In a 1999 paper [11], Gibson defined the biosensor stability as the feature depending mainly on the enzyme immobilization procedures on the biosensor active surface. An operational stability was defined as the preservation of the enzyme catalytic activity, related to the functioning period and the reusability of the biosensor. At the same time, the shelf-stability was demarcated as the amelioration of the enzyme activity when biosensor is stored under well-defined conditions [11]. As stated before, both features are affected by the immobilization procedures, since these can facilitate denaturation processes and the loss of biosensor specificity. Several parameters must be considered in order to evaluate the biosensor stability performance. One of them is V_MAX_, that is the greatest enzymatic rate of conversion of the substrate into products when the enzyme is completely saturated by the substrate [12] and, since this phenomenon occurs when all enzymes have created an enzyme-substrate complex [13], this parameter provides an index of the number of active enzyme molecules present on the biosensor surface [6,14,15]. A second important parameter is K_M_, the Michaelis-Menten constant, which represents the substrate concentration that yields half of the V_MAX_. In particular, K_M_ is related to the affinity of the enzyme for the substrate [6,16] and its deviations have often been linked to variations to substrate/enzyme binding [15]. This parameter results are particularly significant in the development of enzymatic biosensors, because it influences the linear operating range (up to about ½ K_M_): in fact, a wide linear range is linked to a great value of K_M_ [17,18,19]. Additionally, K_M_ is important in order to calculate the linear region slope (LRS), which defines the biosensor sensitivity for the substrate, thus representing the most important analytical parameter [3,18,20].

In the present study, different designs of first-generation glucose and lactate biosensors were tested [6]. The biosensors were based on the same geometry [7] and more, a layer by layer deposition was used, as previously demonstrated [9], and only the nature of the last component creating the containing net, for the layered components, was changed. In the present paper, different biosensor storage conditions were applied. Once manufactured, biosensors were stored at three different temperatures (+4 °C, −20 °C and −80 °C) and then storage stability was assessed by means of full glucose and lactate calibrations, from Day 1 and repeated each week for the entire first month: thus, biosensors were calibrated at 7-14-21-28 days from their construction. Then the biosensors were calibrated once a month over a period of four months. For each design, V_MAX_ and K_M_, and LRS were evaluated.

In general, implantable biosensors don’t have a prolonged stability over time. Therefore, from the temporal point of view, the time of the implant is closely linked, to the time of their construction and calibration because the retention of biosensor enzyme activity can be affected by several factors, such as storage conditions [6].

## 2. Material and Methods

### 2.1. Chemicals and Solutions

All chemicals were analytical reagent grade or higher purity and dissolved in bidistilled deionized water. Ascorbic acid, D-(+)-glucose, L-lactate, glucose oxidase from *Aspergillus Niger* (EC 1.1.3.4), lactate oxidase from *Pediococcus species* (EC 1.1.3.2), albumin from bovine serum (BSA)**,**
*o*-phenylenediamine (OPD), polyethylenimine (PEI), glutaraldehyde (GTA)**,** polyurethane (PU) and tetrahydrofuran (THF) were purchased from Sigma-Aldrich (Milano, Italy). The phosphate-buffered saline (PBS, 50 mM) solution was made using 0.15 M NaCl, 0.05 M NaH_2_PO_4_ and 0.04 M NaOH from Sigma, and then adjusted to pH 7.4. Glucose oxidase (GOx) solution was prepared by dissolving 180 units of enzyme in 10 μL of PBS. Lactate oxidase (LOx) solution was prepared in a similar manner, by dissolving 25 units in 50 μL of PBS. The OPD monomer (250 mM) was dissolved in deoxygenated PBS immediately before use. While concentrated solutions of lactate (1 M) and AA (100 mM) were prepared in water immediately before use, the glucose solution (1 M) was prepared 24 h before its use in order to allow equilibration of anomers [9] and then stored in the fridge at 4 °C. PEI (1%), BSA (2%) and GTA (1%) solutions were prepared in bidistilled water while PU (5%) solution was dissolved in THF. Solutions were kept at +4 °C when not in use. Platinum/Iridium (Pt, 90:10, 125 µm ø) Teflon^®^-insulated wires were purchased from Advent Research Materials (Suffolk, UK). Dry ice was obtained from SAPIO LIFE srl (Porto Torres, Italy)

### 2.2. Design and Construction of Glucose and Lactate Biosensors

The glucose biosensors (Figure 1) were prepared by modifying a previously described procedure [7,9,10].

In brief, at Day 0, a portion of Pt wire of 3 cm was cut and from one edge of the wire a 3 mm portion of Teflon^®^ insulation was eliminated in order to allow welding the bare metal to a connector. Then, a Pt cylinder of 1 mm was obtained on the other edge by eliminating the Teflon^®^ insulation, thus obtaining the active surface of the transducer. As the first step, the electrosynthesis of polyo- phenylenediamine (p-OPD) was carried out at +700 mV vs. Ag/AgCl reference electrode for 15 min after having immersed the working electrodes in a solution containing the OPD monomer (250 mM). The Pt/p-OPD cylinder was immersed by means of quick dips into a solution of PEI (1%) and then glucose oxidase (GOx) to allow enzyme adsorption. After 5 min drying at room temperature, the dipping procedure was repeated other four times. Then two different designs of glucose biosensors were made.

In the first design (glucose biosensor 1, GB1, Figure 1 Panel A) a final layer of PU (5%) was applied by quick dipping the biosensor in the PU solution as previously described [9,21]. The second design (glucose biosensor 2, GB2, Figure 1 Panel B) was obtained by quickly immersing the biosensor in the BSA (2%) solution and, after five minutes of drying, then quick-dipped in the glutaraldehyde (1%) solution. Lactate biosensors were manufactured following the same protocol used for glucose biosensors and naming them lactate biosensor 1 (LB1) when the final net was represented by PU and lactate biosensor 2 (LB2) when the final dip was represented by BSA and GTA (Figure 1 Panel C and D respectively).

The biosensors were then arranged inside a 25 mL sealed and appropriately-modified Falcon^®^ test tube in dry conditions, in order to preserve biosensors from humidity, air and ice but, most of all, in order to prevent the tips of the biosensors from touching the walls

The biosensors were then stored under different conditions, as described in the following paragraph.

### 2.3. Storage Protocol

The storage protocol foresaw the constitution of 16 different groups (n=4 biosensors per group). After manufacture, at Day 0, biosensors were left at room temperature (RT, ≃ 25 ± 1 °C) for 30 min, then two groups were kept in the freezer: one was stored at −20 °C and the second at −80 °C. A third group was placed in the fridge at +4 °C. Each group was calibrated at Day 1, after having been left 30 min to the air, and after calibration it was put back in its own place of storage. The same protocol was followed for the calibrations in the following days, as explained in the paragraph 2.5.

In a second phase of this study, two groups, made up for all biosensor designs, were studied in parallel storing the first group for 28 days at −80 °C and the second one for 26 days at −80 °C and the following two days in a small polystyrene box filled with dry ice. For all biosensors V_MAX_, K_M_ and LRS were evaluated.

### 2.4. Instrumentation and Software

For all electrochemical procedures, a conventional three-electrode cell was utilized consisting of 20 mL of PBS, four biosensors as working electrodes, an Ag/AgCl (3 M) (Bioanalytical Systems, Inc. West Lafayette, Indiana, USA) as reference electrode and a large stainless-steel needle as auxiliary electrode. A four-channel potentiostat (eDAQ Quadstat, e-Corder 410, eDAQ Europe, Warszawa, Poland) was used for all electrochemical experiments.

### 2.5. Biosensors Calibration

All in-vitro calibrations were performed in fresh PBS at room temperature (25 ± 1 °C), following a standard protocol. In brief: at Day 1, the biosensors were removed from the storage container and left at room temperature for 30 min, then connected to the potentiostat and immersed in 20 mL of fresh PBS [9,22]. Constant potential amperometry (CPA) was used for calibrations, by applying an anodic potential of +700 mV vs. Ag/AgCl in order to allow H_2_O_2_ oxidation [23]. After having obtained a stable baseline, a full calibration was performed by adding know volumes of glucose or lactate stock solution ranging from 0 to 140 mM. At the end of calibrations, the biosensors were rinsed in pure water and then stored again in the proper environment after a drying period of 30 min.

For each biosensors’ group, the same protocol was repeated each week for the entire first month. So, biosensors were calibrated at 7-14-21-28 days from their construction. In the next months, from the second to the fourth month, calibrations were performed during the last week in order to respect the monthly cadence.

### 2.6. Statistical Analysis

After calibrations, biosensor currents were plotted versus Glu and Lac concentrations. First, a nonlinear fitting with Michaelis–Menten equation was done on the whole concentration range (0–140 mM) to evaluate V_MAX_ and apparent K_M_ parameters, then linear regressions (slope) were calculated at low concentrations (0–1.0 mM). Recorded currents were expressed in nanoamperes (nA) and given as baseline-subtracted values ± standard error of the mean.

Statistical significance (P values) within each group, compared to Day 1, was evaluated by means of ANOVA by GraphPad Prism 5.02 v software. Regarding the comparison with dry ice group, because new groups of biosensors were manufactured, the statistical significance (P values) was calculated by means of unpaired t-test by means of GraphPad Prism (ver. 5.02) software.

## 3. Results

### 3.1. GB1 Over-Time Performances

In Figure 2 the performance variations of GB1 design, over a period of four months, are shown.

As highlighted in panel A and B, the V_MAX_ modifications over a 28 days (Panel A) and four months (Panel B) period of time of the biosensors GB1 design, when stored at different temperatures, are presented.

It can be deduced from the plots that the storage at +4 °C (red plot) determined a substantial decrease in V_MAX_ in the first 28 days of observation, starting from 308.00 ± 24.54 nA at Day 1 and reaching 241.00 ± 25.14 nA at Day 28. This downward trend was maintained in the following three months. In particular, this decrease was significantly lower, if compared with Day 1 (*p* < 0.01), from the second month up to the fourth, when V_MAX_ was 200.60 ± 17.08 nA (Appendix A).

The storage at −20 °C (green plot), produced, overall, higher values of the V_MAX_s, if compared with +4 °C storage group, also determining a slight upward trend in the first month, which turned out to be significantly higher (*p* < 0.01 vs. Day 1) from Day 7 up to Day 28, when figures passed from 433.00 ± 18.55 nA to 397.56 ± 20.40 nA (Appendix A). The following three months resulted in a minor decrease, when V_MAX_s didn’t result significantly different from Day 1 (Appendix A).

In the −80 °C storage group (blue plot), V_MAX_ displayed a pronounced upward trend from Day 1 up to Day 28, starting from 304.00 ± 15.43 nA at Day 1 and reaching 529.00 ± 26.20 nA at Day 28, which is significantly higher (*p* < 0.01) compared than Day 1. From the second month, V_MAX_s suffered a very slight decrease, resulting in being significantly higher (*p* < 0.01) for the whole duration of the observation period, (Appendix A). Interestingly, the −80 °C storage gave rise to the highest V_MAX_s when compared to the other storage groups.

In panel C and D, the K_M_ modifications over 28 days (Panel C) and four months (Panel D) period of time of the biosensors GB1 design are highlighted.

As expected, in all storage groups, a sustained increase in K_M_s was highlighted in the first month of observation. In particular, the +4 °C storage group (red plot) showed the highest K_M_ values, that turned out to be significantly higher (*p* < 0.01) when compared with Day 1, starting from Day 14 for the whole duration of the monitoring, with a maximum level of 7.35 ± 0.11 mM at Day 60 (Appendix A). In the following three months, a non-significant decrease was observed.

Although the −20 °C and −80 °C groups (green plot and blue plot respectively) showed the same upward trend as +4 °C in the first month, they displayed lower values of K_M_ which turned out to be significantly higher (*p* < 0.01) when compared with the respective Day 1. Moreover, in the following three months, while the −80 °C group showed a variable trend, the −20 °C group showed a steadily rising trend (Appendix A).

In panel E and F, the LRS modifications over 28 days (Panel C) and four months (Panel D) period of time of the biosensors GB1 design are showed. From the plots, an overall sustained decrease can be observed in LRS for +4 °C and −20 °C groups in the first month.

For the +4 °C group (red plot), the decrease turned out to be significant from Day 14 (36.25 ± 2.28 nA mM^−1^) up to Day 28 (28.90 ± 2.17 nA mM^−1^), if matched with Day 1 (*p* < 0.01), as also shown in Appendix A. In the following three months, a slight decrease, although still significant (*p* < 0.01 vs. Day 1), was observed.

Interestingly, the −20 °C group (green plot) presented higher LRS values, if compared with +4 °C group, showing a small, but non-significant, increase between Day 1 and Day 7 (61.06 ± 2.40 nA mM^−1^ and 63.25 ± 3.36 nA mM^−1^ respectively). From Day 14 on, a decrease was monitored, which turned out to be significant from Day 28 (41.25 ± 2.29 nA mM^−1^), and remained as such for the next three months (Appendix A)

Surprisingly, the −80 °C group (blue plot) showed the highest LRS values, if compared with the other groups, and followed a not very pronounced and non-significant downhill trend, notwithstanding a small and not significant increase recorded between Day 1 and Day 7 (60.20 ± 2.41 nA mM^−1^ and 68.17 ± 3.16 nA mM^−1^ respectively), and reaching 52.67 ± 3.31 nA mM^−1^ at Day 120 (Appendix A).

### 3.2. GB2 Over-Time Performances

In Figure 3 the variations of V_MAX_, K_M_ and LRS of GB2 design over 28 days (left inset) and four months period (right inset) of time due to different storage protocols are shown.

In panel A and B, V_MAX_ variations in a range of 28 days (left inset) and of four months (right inset) are displayed.

In the biosensor group stored at +4 °C (red plot), a general decrease was observed over time which turned out to be much more pronounced within the first 28 days, if compared to the next three months, passing from 503.00 ± 28.50 nA at Day 1 to 224.89 ± 19.00 nA at Day 120. Nevertheless, there was a significant increase (*p* < 0.01 vs. Day 1) at Day 7 (636.00 ± 29.00 nA), but from Day 21 onwards a significant decrease was observed (*p* < 0.01 vs. Day 1), as shown in Appendix A.

As shown for the GB1 design, V_MAX_s for −20 °C (green plot) and −80 °C (blue plot) groups gave higher results that the +4 °C group, but, while the −20 °C group underwent a general decrease which was more pronounced from Day 60 onwards (Appendix A), the −80 °C group showed an overall significantly higher rising trend (*p* < 0.01 vs. Day 1), especially at Day 7 and Day 14 (629.03 ± 48.01 nA and 662.02 ± 46.40 nA respectively) and also at Day 120, where V_MAX_ was 623.02 ± 55.04 nA (Appendix A).

In panel C and D, K_M_ variations of above-mentioned design are highlighted. Surprisingly, when biosensors were stored at +4 °C (red plot), a general reduction was monitored, which turned out to be more pronounced in the first 28 days of observation, in particular from Day 14 to Day 28, when K_M_s passed from 1.80 ± 0.11 mM to 1.62 ± 0.13 mM. The only significant variation was recorded at Day 7, when K_M_ increased up to 2.66 ± 0.09 mM, as displayed in Appendix A.

Contrary to what occurred in the GB1 drawing, K_M_s for −20 °C (green plot) and −80 °C (blue plot) groups had higher results than the +4 °C group that also exhibited a variable trend. Actually, the −20 °C group, after an initial significant increase (*p* < 0.01 vs. Day 1) recorded between Day 7 and Day 28 (2.63 ± 0.12 mM and 2.00 ± 0.12 mM respectively), entered a downward trend from the second month, that was not significant if compared to Day 1 (Appendix A). On the other hand, the −80 °C group revealed a constant trend, exhibiting significantly higher values compared to Day 1, reaching an average value of 2.42 mM.

In panel E and F, the over-time variations of LRS in a range of 28 days (left inset) and of four months (right inset) are displayed, according to the different storage protocol.

As displayed in panel E, LRSs in +4 °C group (red plot) suffered a sustained decrease between Day 1 and Day 28 (145.03 ± 4.80 nA mM^−1^ and 78.20 ± 3.70 nA mM^−1^ respectively) and remained almost constant for the following three months, reaching an average value of about 72 nA mM^−1^. The decrease, if compared to Day 1, turned out to be significant (*p* < 0.01) throughout the observation period (Appendix A)

The LRSs in −20 °C (green plot) and −80 °C (blue plot) groups showed higher results than the +4 °C group and showed almost the same downward trend, being even almost overlapping, within the first 28 days of observation. However, in the next three months the behavior changed. While in −20 °C group LRSs continued to decline, being statistically lower (*p* < 0.01 vs. Day 1) from Day 21, in the −80 °C group they remained almost constant at around an average value of 126.02 nA mM^−1^ (Appendix A), although they resulted in a significant decrease (*p* < 0.01 vs. Day 1) from Day 28 on.

### 3.3. LB1 Over-Time Performances

In Figure 4 the over-time variations of LB1 biosensor when subjected to different storage protocols design are showed.

In Panel A and B variations of V_MAX_ are displayed over a period of 28 days (left inset) and four months (right inset).

In the +4 °C group (red plot), a considerable downward tendency was highlighted, which turned out to be more consistent between Day 1 and Day 28 when V_MAX_ passed from 216.01 ± 5.70 nA to 60.25 ± 5.58 nA, and less consistent the following three months, when V_MAX_ reached 11.11 ± 3.60 nA. As shown in Appendix A, the V_MAX_ reduction turned out to be significant (*p* < 0.01 vs. Day 1) from Day 7 throughout the entire period of the observations. Although in the −20 °C (green plot) and −80 °C (blue plot) groups V_MAX_s appeared to be higher, if compared with those of +4 °C group, they also underwent a reducing trend during the first 28 days of study, decreasing by 45% and 27% respectively. In the following three months, while the −20 °C group remained almost constant, settling around an average value of 112.89 nA, the −80 °C underwent a slight increase and reached the value of 179.31 ± 2.72 nA. While in the −20 °C group the variations were significant (*p* < 0.01 vs. Day 1) already from Day 7, in the −80 °C group significant variations (*p* < 0.01 vs. Day 1) were obtained from Day 14, as displayed in Appendix A.

In panel C and D, the over-time variations of K_M_ in a range of 28 days (left inset) and of four months (right inset) are displayed, according to the different storage protocol.

In the +4 °C group (red plot), K_M_ did not show a predictable trend, as it underwent an initial significant increase of about 90% (*p* < 0.01 vs. Day 1), between Day 1 and Day 28, and then decreased in the following three months, resulting not significantly different than Day 1 (*p* < 0.01) (Appendix A).

Although the group stored at −20 °C showed slightly higher values of K_M_ if compared to −80 °C group values, both exhibited a similar increasing trend, but which tended to be constant in the −20 °C group, while the −80 °C group turned out to fluctuate more. In both groups, the only significant variation occurred between the third and the fourth month of observation (Appendix A).

In panel E and F, the over-time variations of LRS in a range of 28 days (left inset) and of four months (right inset) are displayed, according to the different storage protocol.

As shown in the scatter plot, the +4 °C and −20 °C groups underwent a substantial decline, which was more marked between Day 7 and Day 28, but quite substantial even in the next three months. However, LRSs in the −80 °C group surprisingly remained almost constant, settling around a value of about 36.20 nA mM^−1^. The +4 °C group suffered the highest diminution, passing from to 32.05 ± 1.06 nA mM^−1^ at Day 1 to 1.79 ± 0.14 nA mM^−1^ at Day 120. Variation in +4 °C and −20 °C groups was significant (*p* < 0.01 vs. respective Day 1) from Day 7 on, as shown in Appendix A, while LRS modifications monitored in −80 °C group turned out to be non-significant *p* < 0.01 vs. respective Day 1) (Appendix A).

### 3.4. LB2 Over-Time Performances

In Figure 5, over-time variations of LB2 biosensor design are showed.

In panel A and B, V_MAX_ variations are displayed. While +4 °C (red plot) and −20 °C (green plot) groups suffered a dramatic V_MAX_ decrease over time (more pronounced in the first month), the −80 °C the group not only proved to possess the highest V_MAX_ values, but they only remained almost stable around a value of about 28 nA, changing significantly only from Day 28 onwards (*p* < 0.01 vs. respective Day 1), as shown in Appendix A. In particular, in the +4 °C group V_MAX_ suffered a decrease over time of about 95%, while in the −20 °C showed a decrease of about 80%.

In panel C and D, K_M_ variations are highlighted. This parameter showed a non-predictable behavior for the all considered group, remaining relatively more stable in −20 °C and −80 °C groups (Appendix A).

In panel E and F, the over-time variations of LRS in a range of 28 days (left inset) and of four months (right inset) are displayed, according to the different storage protocol. All the three groups showed the same downward tendency over time. But, while the +4 °C group LRS suffered a decline of more than 90% (passing from 25.7 ± 0.96 nA mM^−1^ at Day 1 to 1.56 ± 0.10 nA mM^−1^ at Day 120), in the −20 °C group a diminution occurred of about 53%. Instead, the −80 °C group, which moreover was revealed to have the highest LRS values among the three groups, suffered an over-time decrease of only about 30%. Moreover, as shown in Appendix A, while in +4 °C and −20 °C groups all the variations had significant changes from Day 7 onward, in the −80 °C group the modifications were significant only from Day 21 onward.

### 3.5. Effect of Dry Ice Storage

In this group of biosensors, data obtained for −80 °C storage were no different from those of previous groups, obtained from calibrations performed after freezing-thawing processes.

Thus, in Table 1 the effects of the storage in dry ice, following the storage at −80 °C, of the all biosensor designs in terms of V_MAX_, K_M_ and LRS are shown. Variations of the above-mentioned parameters were compared to those obtained from the same designs stored only at −80 °C.

As shown in Table 1, GB1 design, when stored only at −80 °C, displayed comparable parameters as described in paragraph 3.1, while the 48 h-storage in dry ice determined a non-significant variation of the parameters considered. In fact, V_MAX_ and LRS suffered a diminution of about 8% and 6% respectively, while K_M_ increased by about 8%. However, these variations were not significant if compared to the corresponding biosensor group stored at −80 °C.

About GB2 design, in the group stored also in dry ice the trend of the variations of the three parameters was similar. In fact, for V_MAX_ and LRS a reduction of about 12% and 18% respectively occurred, while an increase of about 38% was recorded for K_M_. Even in this group, there were no statistically significant differences in the parameters.

Regarding LB1 design, the dry ice storage did not cause any significant variation in the parameters, if compared to the group subjected to freezing only at −80 °C. Actually, while K_M_ suffered an increase of about 27%, a decrease of 13% and 27% occurred for V_MAX_ and LRS respectively.

In LB2 design, in contrast to the other groups, the 48 h-storage in dry ice determined significant differences (*p* < 0.05) in all the observed parameters, if compared with the corresponding biosensor group stored only at −80 °C. In fact, while V_MAX_ and K_M_ underwent a considerable diminution of about 30% and 40% respectively, while for K_M_ an increase of +120% occurred.

## 4. Discussions and Conclusions

As can be deduced from the data, exposing biosensors to different storage temperatures determined diverse behaviors regarding the different parameters. In all the studied designs, a temperature-dependent performance was observed for V_MAX_s. In fact, while at +4 °C this parameter suffered a general decrease, at −20 °C the phenomenon was partially attenuated, producing higher values compared to +4 °C data. Moreover, the storage at −80 °C determined higher V_MAX_ values, if compared with the other two groups, and also determined a definitively better stability level over time. This phenomenon not only revealed that at very-low storage temperatures GOx and LOx enzymes preserved their activity, but also that the number of active molecules on the biosensor surface was, at least formally, increased [6,14,15]. A similar behavior was observed for the LRS parameter. Actually, the +4 °C storage didn’t preserve the sensitivity of biosensors, which decreased over time for all studied designs, while the parameter was conserved, and indeed increased, when biosensors were exposed to −20 °C and −80 °C. As previously demonstrated, LRS varies dependently on V_MAX_ [14,18,20] so the −80 °C storage produced an interesting over-time LRS stability in the considered range of time of 120 days. Moreover, the presence of PU determined higher values of LRS, if compared with GTA+BSA design, and kept them higher throughout the considered range of time. As expected, regardless of the conservation protocol, the presence of PU in the design determined higher values of K_M_ [21,23], due to its thickness of about 7 μm, thus creating an apparent loss in substrate affinity [9]. Another interesting point to be investigated and better elucidated is the eventual role of PU membrane in the protection of enzyme molecules during freeze-thawing processes. Also, considering the results obtained with the groups stored in dry ice, PU proved to be more effective in the retention of biosensor performances when compared with GTA + BSA net, which was not able to safeguard the biosensors over-time features.

Besides, in our laboratory experience, the solubilized enzymes are stored in aliquots at −80 °C when not in use, resulting in a long-lasting stability and reproducibility (data not shown). In the current study, the immobilized enzymes layered on the biosensor surface stabilized over-time and, in some cases, enhanced their performances as well as the solubilized and aliquoted enzymes.

In addition, the role of PEI deserves to be investigated in more depth, not only just about the already highlighted enhancement of enzyme activity [14,15] but also in the protection of molecules during freeze-thaw processes [24,25,26,27]. In fact, in a previous publication [27], it has been demonstrated that the occurrence of PEI is able to influence the microenvironment of the enzyme, interacting not only with proteins but also with the ionic substrates and products. Moreover, our use of PEI at high concentrations during freeze-thawing processes helped to retain activity and to provide a higher protection for enzyme molecules, avoiding some denaturizing processes during subunits’ dissociation and unfolding in the freezing state.

At this stage, we are unable to define what contribution is to be attributed to the various components of the biosensor in the over-time stabilization and enhancement, or regarding the protection of the enzyme molecules from the ice crystals eventually formed during freezing and, moreover, in the absence of cryoprotectants. In fact, it has been demonstrated that small ice crystals can expose proteins to damage and also thawing can cause a further impairment due to recrystallization [24] or rehydration processes [25]. In fact, it is well known that the presence of some compounds, as glycerol, are able to stabilize enzymes [3] and protect them during freezing [24]. Thus, at the moment some experiments that provide the introduction of glycerol in the studied biosensor design are in progress, in order to evaluate what impact this component could have in the over-time stability of biosensors, when stored at very low temperatures.

Having observed that 48 h in dry ice, following 26 day of −80 °C storage, didn’t determine any significant variations in biosensors’ performance, this result opens up new interesting perspectives to dissociate the moment of the construction and the storage/delivery of biosensors, hypothesizing to exploit the low temperatures to stabilize the sensors also during the transport. Those findings, although valid for both designs with GOx, weren’t valid for the biosensor loading LOx and GTA + BSA, which showed the worst performances during very low temperature storage. So, the study on low-temperature storage/delivery must be done in relation not only to the used enzyme but also to the biosensor design.

In view of the considerations expressed above, and because no publication in the literature presently discusses the effect of freezing on the performance of biosensors, the aim of the present study was to add a further characterization of some biosensors’ designs, already published by our research group [9,10], so that we would be able to respond to the need to separate the moment of the biosensor manufacture and its implantation in animal models’ brains. Moreover, these observations could be of some importance in the eventuality that the biosensors must be implanted, following an opportune storage and an appropriate transport, in a different place from their manufacture, an aspect that could be of some importance in the eventual industrial production of biosensors and their relative commercialization.

For these reasons, the present work had the further objective of investigating the effect of storing biosensors in dry ice, for a short period of time, simulating the hypothetical necessity of sending the biosensors from the manufacturing facility to another laboratory, after a storing period at −80 °C.

Since the method opened by this study seems to be quite promising, it deserves to be progressed further in future studies.

## Figures and Tables

**Figure 1 sensors-19-00422-f001:**
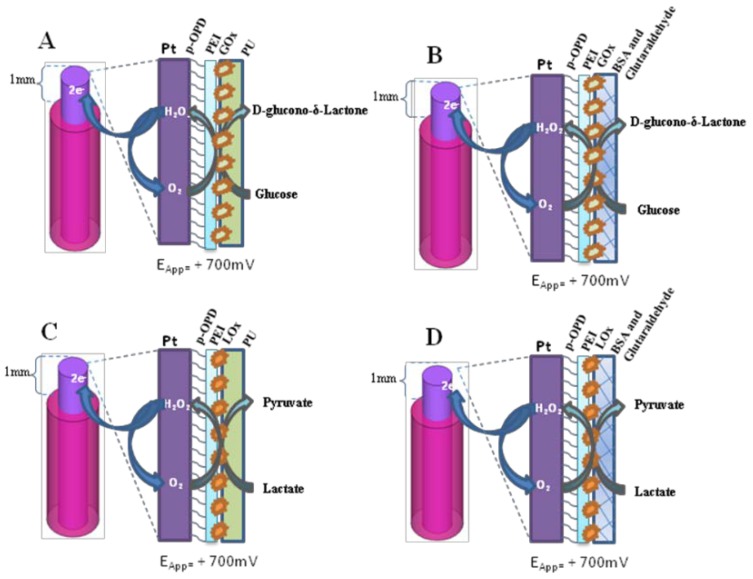
Graphical representation of two main designs of glucose and lactate biosensors used in this study: GB1 (Panel **A**): Pt_c_/PPD/[PEI(0.5%)-GOx]_5_/PU (5%); GB2 (Panel **B**): Pt_c_/PPD/[PE(0.5%)-GOx]_5_/GTA(1%)-BSA(2%); LB1 (Panel **C**): Pt_c_/PPD/[PEI(0.5%)-LOx]_5_/PU (5%); LB2 (Panel **D**): Pt_c_/PPD/[PEI(0.5%)-LOx]_5_/ GTA(1%)-BSA(2%); Pt_c_: Pt cylinder 1 mm long, 125 μm diameter; GOx: glucose oxidase; LOx: lactate oxidase PPD: poly-ortho-phenylenediamine; PEI: polyethyleneimine; PU: polyurethane; GTA: glutaraldehyde; BSA: bovine serum albumin. The subscript “x” represents the number of dip−evaporation deposition stages and in brackets the concentration of the component.

**Figure 2 sensors-19-00422-f002:**
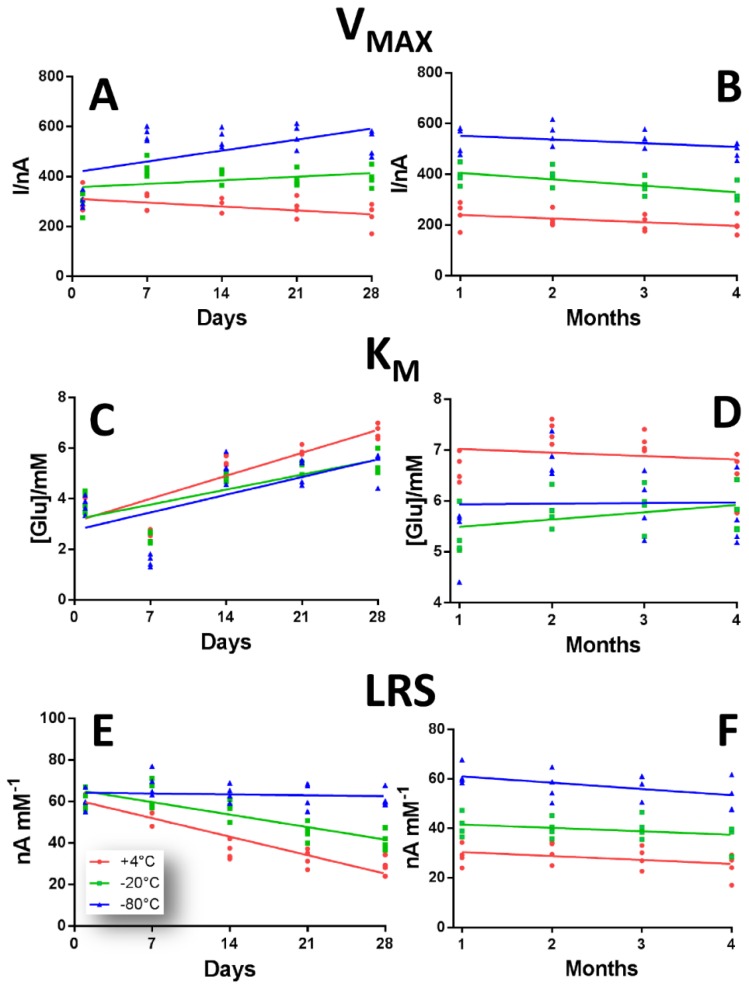
Scattering plot describing the variations of V_MAX_ (Panel **A**,**B**), K_M_ (Panel **C**,**D**) and LRS (Panel **E**,**F**), in a range of 28 days (left inset) and of four months (right inset) of GB1 design Pt_c_/PPD/[PEI (0.5%)-GOx]_5_/PU (5%) when stored at +4 °C (red plot), −20 °C (green plot) and −80 °C (blue plot).

**Figure 3 sensors-19-00422-f003:**
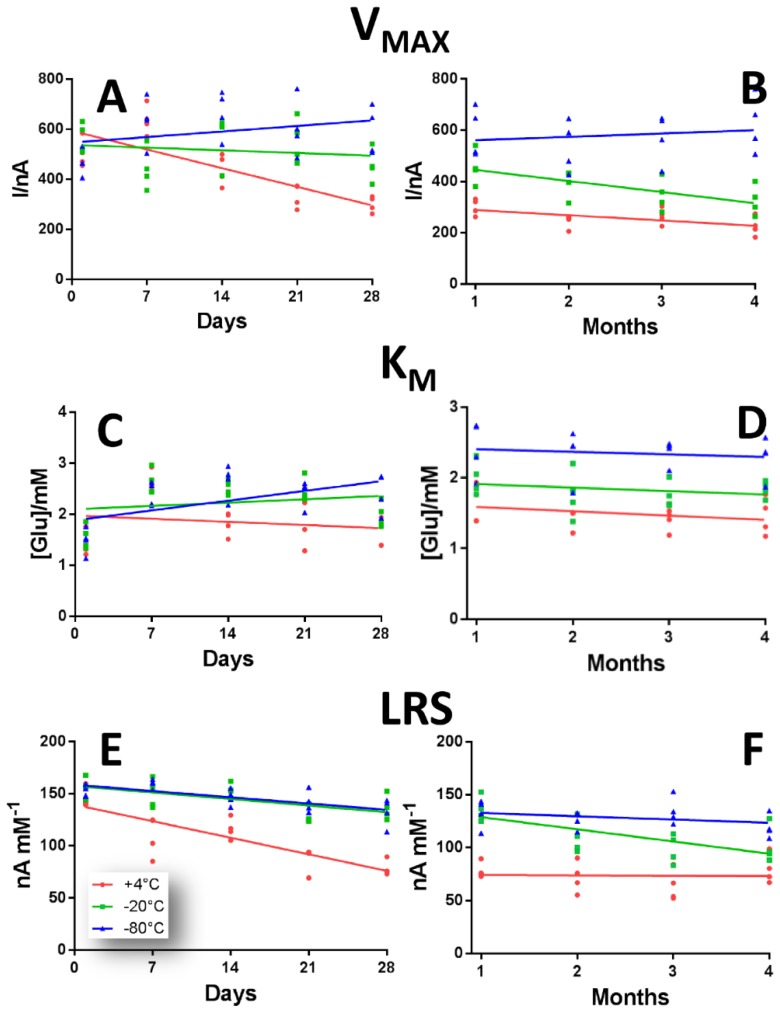
Scattering plot describing the variations of V_MAX_ (Panel **A**,**B**), K_M_ (Panel **C**,**D**) and LRS (Panel **E**,**F**), in a range of 28 days (left inset) and of four months (right inset) of GB2 design Ptc/PPD/[PEI (0.5%)-GOx]5/ GTA(1%)-BSA (2%) when stored at +4 °C (red plot), −20 °C (green plot) and −80 °C (blue plot).

**Figure 4 sensors-19-00422-f004:**
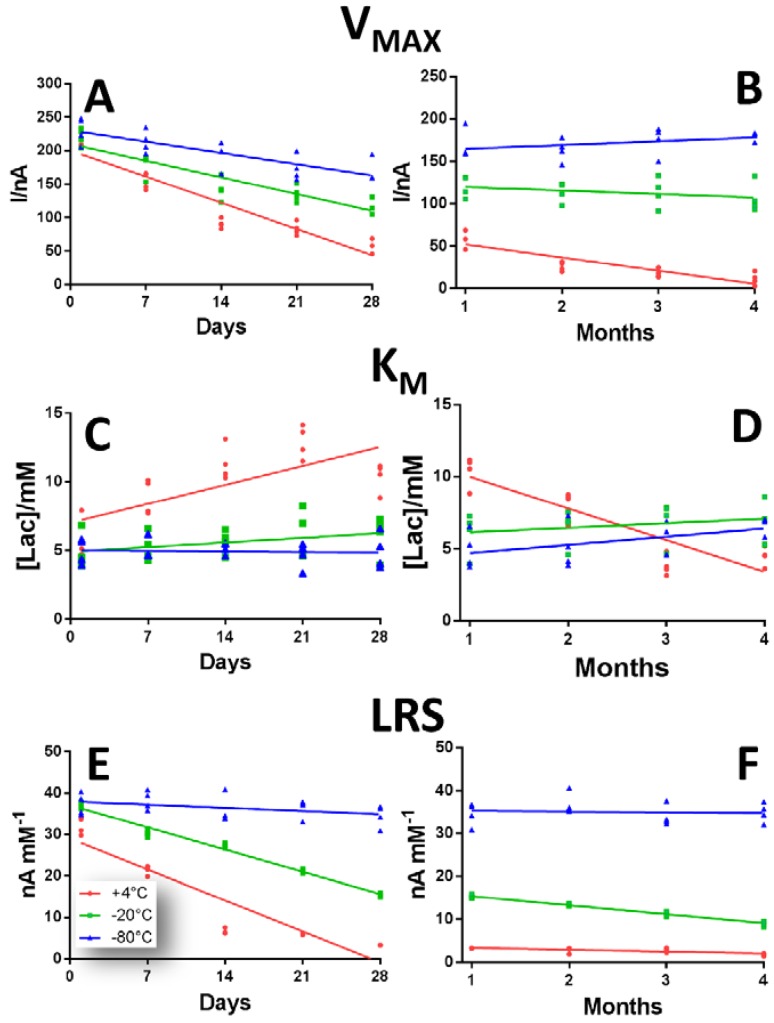
Scattering plot describing the variations of V_MAX_ (Panel **A**,**B**), K_M_ (Panel **C**,**D**) and LRS (Panel **E**,**F**), in a range of 28 days (left inset) and of four months (right inset) of LB1 design Ptc/PPD/[PEI (0.5%)-LOx]_5_/ GTA(1%)-BSA (2%) in the defined range of 120 days, when stored at +4 °C (red plot), −20 °C (green plot) and −80 °C (blue plot).

**Figure 5 sensors-19-00422-f005:**
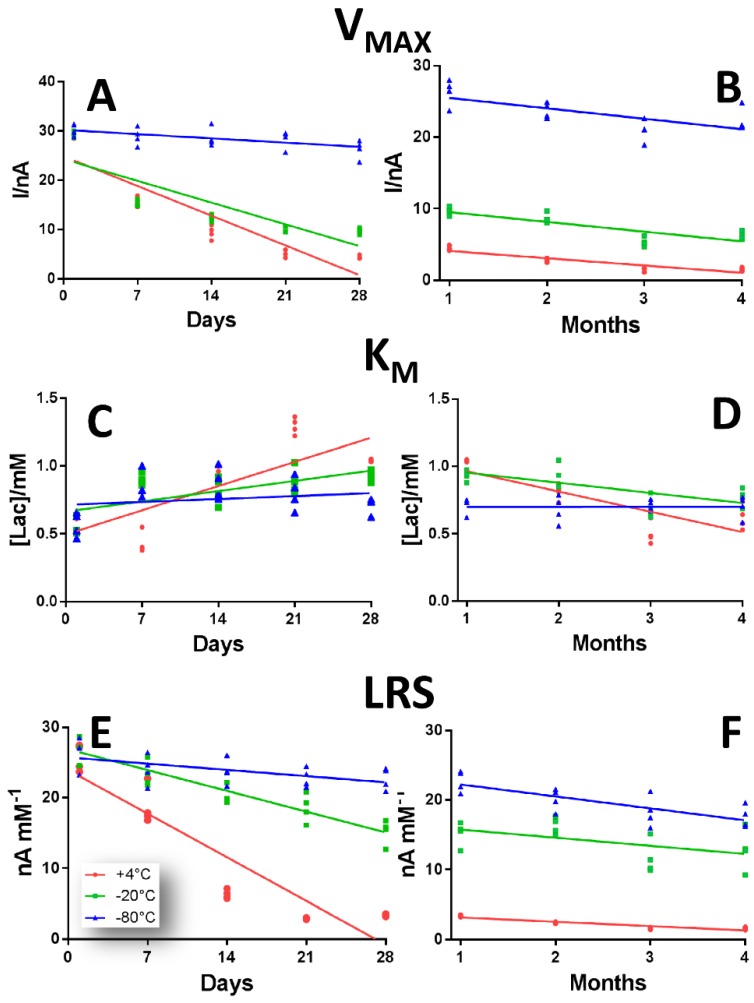
Scattering plot describing the variations of V_MAX_ (Panel **A**,**B**), K_M_ (Panel **C**,**D**) and LRS (Panel **E**,**F**), in a range of 28 days (left inset) and of four months (right inset) of LB2 design Ptc/PPD/[PEI (0.5%)-LOx]_5_/ GTA(1%)-BSA (2%) in the defined range of 120 days, when stored at +4 °C (red plot), −20 °C (green plot) and −80 °C (blue plot).

**Table 1 sensors-19-00422-t001:** Biosensor kinetic parameters obtained after different storage conditions. Four groups of biosensors (*n* = 4 *per* design) were stored at −80 °C for 28 days and calibrated immediately after stabilization in PBS at RT. Further four groups of biosensors were used for simulating the transport in dry ice by storing the biosensors at −70 °C for 48 h after a period of 26 days at −80 °C. * = *p* < 0.05 vs. the corresponding biosensor group stored at −80 °C for 28 days.

Biosensor Design	V_MAX_ (nA)		K_M_ (mM)		LRS (nA mM^−1^)	
	(28 days at −80 °C)	(26 days at −80 °C plus 48 h at −70 °C)	(28 days at −80 °C)	(26 days at −80 °C plus 48 h at −70 °C)	(28 days at −80 °C)	(26 days at −80 °C plus 48 h at −70 °C)
**GB1**	529.03 ± 26.21	487.10 ± 33.5	5.33 ± 0.31	5.74 ± 0.41	61.24 ± 2.11	57.81 ± 3.61
**GB2**	592.01 ± 47.90	523.51 ± 52.7	2.42 ± 0.19	3.35 ± 0.33	132.11 ± 6.80	109.10 ± 8.91
**LB1**	168.02 ± 8.60	139.11 ± 10.20	4.89 ± 0.63	6.20 ± 0.86	32.24 ± 1.35	23.61 ± 3.21
**LB2**	26.30 ± 0.69	18.50 ± 2.42 *	0.79 ± 0.06	1.74 ± 0.12 *	22.72 ± 0.78	13.50 ± 1.31 *

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
