# Peer review of "Low-Temperature Storage Improves the Over-Time Stability of Implantable Glucose and Lactate Biosensors"

_sensors, 2019, doi:10.3390/s19020422_

Reviewer 1 Report

The quality of the manuscript has not been significantly improved.

Author Response

REVIEWER #1

The manuscript has been modified responding to most of the remarks. I however invite the authors to reconsider the histograms for data visualization, as no regular interval between the measurements occurs and they are deceiving,

In order to meet the reviewer's request, authors changed data representation from bar chart plots into scattering plots. In particular, to express data in a regular interval of time, two different graph types are produced for each design: one showing the variations of the considered parameters within the first month (from Day 1 to Day 28) and the second displaying the variations between the first and the fourth month. The relative legends were modified, as well as the paragraphs from 3.1 to 3.4 explaining the plots. Moreover, for a more exhaustive comprehension of data, bar charts have been collected in the Supplementary Materials file.

 Moreover, the increase in Vmax does not reflect the number of active sites (a constant) but its increase just reveals a faster reaction rate, i.e. due to molecular rearrangements? conformational changes? surface crowding? Knowning the amount of enzyme immobilized, an apparent kcat could be calculated to reveal the catalytic efficiency and better compare the biosensors. The curves used for calculating km and Vmax should also be added as supplementary material or panel in a figure.

In order to answer to the reviewer remarks about VMAX, in literature it is possible to find numerous published papers affirming that the VMAX, which in some papers is also expressed as JMAX and in others as IMAX, reflects the enzyme loading on the biosensor surface. The authors reported only a few:

1) “Different values of Jmax, determined under the same conditions, reflect differences in the amount of active enzyme on the surface (McMahon CP, Rocchitta G, Serra PA, Kirwan SM, Lowry JP, O'Neill RD, Analyst. 2006 131(1):68-72; Mc Mahon et al Biosens Bioelectron. 2007 Feb 15;22(7):1466-73).

2) “variations in JMAX may occur which are due to changes in biosensor HP sensitivity, enzyme loading or enzyme activity” (Ford et al Sensors (Basel). 2016 Sep 23;16(10). pii: E1565)

3) “In fact, VMAX values had a surprising upward trend up to Day 15, indicating that the number of active molecules on the transducer surface was formally increased during the days. (Rocchitta et al Chemosensors 2018, 6, 58; doi:10.3390/chemosensors6040058).

4) “Different values of JMAX, determined under the same conditions, reflect differences in the amount of active (not total) enzyme on the surface” (Rothwell et al Sensors 2010, 10(7), 6439-6462).

5) “IMAX is a measure of the amount of active enzyme molecules on related biosensor surfaces” (Secchi O, Zinellu M, Spissu Y, Pirisinu M, Bazzu G, Migheli R, Desole MS, O'Neill RD, Serra PA, Rocchitta G, Sensors 2013, 13, 9522-9535).

6) “We have observed a dramatic decrease in IMAX (≥2fold, p≤0.001), suggesting a decrease in the amount of active enzyme. (C.A Cordeiro et al Biosensors and Bioelectronics, Volume 67, 15 May 2015, Pages 677-686)

Besides, as previously published (O'Neill, R.D., Rocchitta, G., McMahon, C.P., Serra, P.A., Lowry, J.P, Trends in Analytical Chemistry, Vol. 27, No. 1, 2008), the activity of an immobilized enzyme on the biosensor surface depends on the amount of the active molecules (defined as density) expressed as [E] but also on the value of the catalytic rate constant, k2, also known as kcat

So, as the enzyme activity is calculated by k2[E], for implantable biosensors is not possible to obtain this value because these two terms are rarely separable, especially when dip evaporation is used to load enzyme on the biosensor surface ([E] is not possible to known independently), as in case of our biosensor procedures.

So, the only parameter we have to estimate enzyme activity is VMAX, since is not possible to calculate the exact amount of enzyme loaded on the biosensor surface.

Actually, as expressed in the previous revision, the authors are unable to define the causes of the VMAX increase when biosensors are stored at very low temperatures. As suggested by the reviewer, probably the phenomena underlying this evidence could be a molecular rearrangement or some conformational changes, since there are no covalent bonds among the enzyme molecules.

Reviewer 2 Report

The manuscript titled “Low-temperature storage improves the over-time stability of implantable glucose and lactate biosensors” studied the effect of storage temperature on the over-time stability of glucose and lactate biosensors by analyzing three parameters of VMAX, KM, and LRS. The study is of importance particularly to production and practical applications. It could be accepted for publication in sensors provided the authors considering the following comments:

1.    It’s surprising that the low temperature storage can improve the performance of the studied sensors in comparison with those fresh prepared (Day 1). As a matter of fact, the best result one can expect for the activity of an enzyme stored under the appropriate conditions is as good as that of the fresh enzyme. The authors should give a reasonable explanation.

2.    The caption in Figure 1 is not clear, for instance, the abbreviations of GB1, GB2, LB1, LB2 are not marked with the corresponding sensors in figure, respectively.

3.    The captions in Figure 2-5 are misleading: For example, “VMAX (Panel A, white bars)” should be “VMAX (Panel A)”. The white bars, light gray bars, and dark gray bars in each panel represented the corresponding parameters at 4, -20, and -80 oC, respectively.

4.    Page 14. “Actually, 4 the +4°C storage didn’t preserve the sensitivity of biosensors”, should the first “4” be deleted?

5.    There are many small errors in this manuscript, for example:

Figure 1: Space is missed between () and text;

Figure 3: VMAX, KM MAX and M should be in subscript format; .

Reference7: 2009 should be in bold.

Author Response

REVIEWER #2

The manuscript titled “Low-temperature storage improves the over-time stability of implantable glucose and lactate biosensors” studied the effect of storage temperature on the over-time stability of glucose and lactate biosensors by analyzing three parameters of VMAX, KM, and LRS. The study is of importance particularly to production and practical applications. It could be accepted for publication in sensors provided the authors considering the following comments:

 It’s surprising that the low temperature storage can improve the performance of the studied sensors in comparison with those fresh prepared (Day 1). As a matter of fact, the best result one can expect for the activity of an enzyme stored under the appropriate conditions is as good as that of the fresh enzyme. The authors should give a reasonable explanation.

 As stated in the Discussion and conclusion paragraph “At this stage, we are unable to define what contribution is to be attributed to the various components of the biosensor in the over-time stabilization and enhancement, or regarding the protection of the enzyme molecules from the ice crystals eventually formed during freezing and, moreover, in absence of cryoprotectants. In fact, it has been demonstrated that small ice crystals can expose proteins to damage and also thawing can cause a further impairment due to recrystallization [24] or rehydration processes [25]. In fact, is well known that the presence of some compounds, as glycerol, are able to stabilize enzymes [3] and protect them during freezing [24].”

As quoted in the text, VMAX is an index of the number of active molecules on the surface of the biosensor (“this parameter provides an index of the number of active enzyme molecules present on the biosensor surface [6, 14, 15]”). Of course, no new enzyme molecules were added during storage, but the freezing-thawing processes probably allowed more molecule to be oriented in the space more favorably to the interaction with the substrate. We would like also to mention that enzymes, loaded on the biosensor surface by means of quick-dipping procedures, are not covalently immobilized, so they are able to reorient.

Moreover, we have also pointed out the role of PEI. In fact, in above-mentioned paragraph, we stated that “Moreover, the use of PEI at high concentrations, how those used by us, during freeze-thawing processes was able to retain activity and to provide a higher protection for enzyme molecules, avoiding some denaturizing processes as subunits’ dissociation and unfolding in the freezing state. Probably, those two events underlay the “at least apparent” increase of the number of active molecules on the biosensor surface.

  2.    The caption in Figure 1 is not clear, for instance, the abbreviations of GB1, GB2, LB1, LB2 are not marked with the corresponding sensors in figure, respectively.

We would like to thank the reviewer for having highlighted this discrepancy. Actually, the legend of Figure 1 has been changed.

 3.    The captions in Figure 2-5 are misleading: For example, “VMAX (Panel A, white bars)” should be “VMAX (Panel A)”. The white bars, light gray bars, and dark gray bars in each panel represented the corresponding parameters at 4, -20, and -80 oC, respectively.

We appreciate the suggestion of the reviewer. Actuallty, the legends of Figure 2-5 have been changed by virtue of the changes requested by the reviewer 1.

4.    Page 14. “Actually, 4 the +4°C storage didn’t preserve the sensitivity of biosensors”, should the first “4” be deleted?

We appreciate the reviewer’s observation. Actually, the mistyping has been corrected.

 5.    There are many small errors in this manuscript, for example:

 Figure 1: Space is missed between () and text;

 Figure 3: VMAX, KM, MAX and M should be in subscript format; .

 Reference7: 2009 should be in bold.

 We appreciate the reviewer attention. All the mistypings’ have been corrected.

Reviewer 3 Report

The work " Low-temperature storage improves the over-time stability of implantable glucose and lactate biosensors" aimes to evaluate the kinetic parameters of two different designs for glucose and lactate biosensor, studied under different storage conditions (+4, -20 and -80 °C). Despite, the work appears to be performed thoroughly, the study is lacking of information addressing previous works with the study of biosensors under unusual storage conditions, is there another study that tested those temperatures? Which were the analytes detected, which were the matrices? How were their performance in comparison? It is recommended to restructure the introduction to highlight the new insights or substantial innovations of the study. Minor spell check is also required and the correction of possible typos (missing spaces e.g. line 137, line139 have an extra space in the word “been”)

Author Response

REVIEWER #3

The work " Low-temperature storage improves the over-time stability of implantable glucose and lactate biosensors" aimes to evaluate the kinetic parameters of two different designs for glucose and lactate biosensor, studied under different storage conditions (+4, -20 and -80 °C). Despite, the work appears to be performed thoroughly, the study is lacking of information addressing previous works with the study of biosensors under unusual storage conditions, is there another study that tested those temperatures? Which were the analytes detected, which were the matrices? How were their performance in comparison? It is recommended to restructure the introduction to highlight the new insights or substantial innovations of the study. Minor spell check is also required and the correction of possible typos (missing spaces e.g. line 137, line139 have an extra space in the word “been”).

Authors appreciate reviewer’s comment on the manuscript, but they would like to put the referee aware of the fact that, according to our research, we haven’t found any papers related to the storage of amperometric and platinum-based biosensors at such low temperatures: the authors would undoubtedly have cited them if they were present in literature.

The authors referred to this topic in the “Discussion and conclusions paragraph” where they stated that “In view of the considerations expressed above, and because no publication is present in the literature about the effect of freezing on the performance of biosensors, the aim of the present study was to add a further characterization of some biosensors’ designs, already published by our research group [9,10], so that we would be able to respond to the need to separate the moment of the biosensor manufacture and its implantation in animal models’ brain”. The modified manuscript was submitted again just before the referee's comment arrived. So, we apologize if the referee could not read this change. Probably, even the typos from him/her highlighted have been corrected.

Reviewer 4 Report

The manuscript presented an interesting work in the operation and shelf stability of the biosensor.This work is well prepared and can be considered to publish after a minor revision. My comments are listed below.

In the section of “ Discussion and conclusion”, the conclusions are not clear. The conclusions for the whole work should be strengthen.  

Author Response

REVIEWER #4

The manuscript presented an interesting work in the operation and shelf stability of the biosensor. This work is well prepared and can be considered to publish after a minor revision. My comments are listed below.

 In the section of “ Discussion and conclusion”, the conclusions are not clear. The conclusions for the whole work should be strengthen. 

 We appreciate the referee’s annotation, and, in accepting this, the paragraph “Discussion and conclusion” was amended as follows:

 In view of the considerations expressed above, and because no publication is present in the literature about the effect of freezing on the performance of biosensors, the aim of the present study was to add a further characterization of some biosensors’ designs, already published by our research group [9,10], so that we would be able to respond to the need to separate the moment of the biosensor manufacture and its implantation in animal models’ brain. Moreover, these observations could be of some importance in the eventuality that the biosensors must be implanted, following an opportune storage and an appropriate transport, in a different place from their manufacture, an aspect that could be of some importance in the eventual industrial production of biosensors and their relative commercialization.

For these reasons, the present work had the further objective of investigating the effect of storing biosensors in dry ice, for a short period of time, simulating the hypothetical necessity of sending the biosensors from the manufacturing facility to another laboratory, after a storing period at -80°C.

Since the way opened by this study seems to be quite promising, it deserves to be deepened in future studies.

Round  2

Reviewer 1 Report

The manuscript will be of interest for sensor manufacturers.